# Resilience, Coping Self-Efficacy, and Posttraumatic Stress Symptoms among Healthcare Workers Who Work with Refugees and Asylum Seekers in Greece

**DOI:** 10.3390/bs14060509

**Published:** 2024-06-20

**Authors:** Bita Ghafoori, Sofia Triliva, Panagiota Chrysikopoulou, Andreas Vavvos

**Affiliations:** 1Department of Advanced Studies in Education and Counseling, California State University Long Beach, 1250 Bellflower Blvd., Long Beach, CA 90840-2201, USA; 2Department of Psychology, Faculty of Social Sciences, University of Crete, Gallos Campus, 74150 Rethymno, Greece; triliva@uoc.gr (S.T.); panagiotachrysikopoulou@gmail.com (P.C.); vavvosandreas@gmail.com (A.V.)

**Keywords:** healthcare workers, refugees, PTSD, resilience, coping

## Abstract

Due to occupational exposure to potentially traumatic events, health care workers (HCWs) may be at risk of developing posttraumatic stress (PTS) symptoms or probable posttraumatic stress disorder (PTSD). This study examined probable PTSD, coping, and resilience among national HCWs working in Greece. A total of 17.9% of the sample of participants (N = 112) met the screening criteria for probable PTSD. Logistic regression models were constructed to assess if trauma coping self-efficacy (CSE) and resilience predicted probable PTSD, and the results indicated that lower trauma CSE significantly predicted probable PTSD in unadjusted models (OR = 0.89, 95% CI, 0.82, 0.96, *p* < 0.01) and adjusted models (OR = 0.90, 95% CI, 0.83, 0.97, *p* < 0.01). Our study findings suggest that organizations that employ HCWs may support their workers through ongoing screening, assessment, and training that enhances coping self-efficacy.

## 1. Introduction

The number of refugees and asylum seekers worldwide is a growing international concern. By mid-2023, 36.4 million refugees and 110 million forcibly displaced people were counted worldwide [1]. Refugees have increased in many countries within the European Union (EU), and the number of refugees that have entered Greece since 2014 has exceeded one million [1]. Greece has hosted over 150,000 refugees and asylum-seekers thus far, and approximately 50,000 refugees have remained in Greece indefinitely [1]. In 2021, the Greek government began managing the mental health care of refugees due to the closure of the Emergency Support to Integration and Accommodation (ESTIA) program that was developed by the UNHCR. The care of the refugee and asylum seeker populations was assigned to Greek national healthcare workers (HCWs). The HCWs were contracted by humanitarian organizations (NGOs) and funded via renewable contracts that were short-term in nature.

First responders’ mental health outcomes have become the focus of additional research in the last 20 years [2]. Healthcare workers have been identified as particularly at risk of developing posttraumatic stress (PTS) symptoms or probable posttraumatic stress disorder (PTSD) due to the stressful work situations to which they are exposed and which may include caring for traumatized populations and the frequent witnessing of death and trauma [2,3]. The Diagnostic and Statistical Manual of Mental Disorders (DSM)-5 [4] states that “experiencing repeated or extreme exposure to aversive details of the traumatic event(s)” can be regarded as potentially traumatic events, and past research suggests that such circumstances may include HCWs repeated exposure to details of traumatic events [2]. Examples of exposure to stressful life experiences or potentially traumatic events that HCWs may encounter have been documented in the wider literature and may include overexposure to life threatening illnesses [5], safety concerns [6], low decision latitude and control [7], unlawful behavior by their clients and/or toward their clients [8], sudden unexpected deaths, deaths caused by suicide or homicide, or being witness to a dead body [6].

National healthcare workers who work with refugees may experience personal, vicarious, communal, episodic, or continuous trauma as part of their occupational role [9]. Work-related traumatic stress exposure, particularly in health care, has been widely documented [10,11]. Four past studies focused specifically on refugee populations and several past studies focused on homeless populations have found that listening to clients’ stories of traumatic experiences and suffering corresponded to a stressful or potentially traumatic occupational demand [6,7,8,12,13,14,15]. Past research on social workers, mental health professionals, nurses, and doctors has found a higher risk of PTSD symptoms among healthcare professionals who worked with trauma-exposed populations [16,17,18]. According to a recent study by Pentaraki and Dionysopoulou [19], social workers in Greece who work with refugees face daily material insecurity as a result of precarious employment and temporary contracts, as well as severe economic and interpersonal challenges, fatigue, and a higher risk of mental health issues. Although HCWs who experience occupational exposure to traumatized refugees may be at high risk of psychological impairment or PTSD symptoms, the literature has shown mixed results and some research suggests that HCWs may not be negatively impacted and instead may report resilience or positive coping [20].

One conceptual framework that may be used to further understand psychological adaptation among HCWs who have been exposed to occupational trauma is social cognitive theory [21]. This theoretical framework describes coping self-efficacy (CSE), defined as an individual’s perceived ability to cope effectively with stressful life events [21]. A growing body of literature has described trauma-related CSE, considered to be a measure of psychological adaptation and defined as an individual’s perception that he or she can manage potentially traumatic events, including distress symptoms, and maintain functioning [21]. Coping self-efficacy is influenced by both personal and secondary experiences and physiological arousal, and CSE helps guide adaptive coping processes through self-evaluation [22]. Trauma-related CSE has been found to be inversely related to PTSD symptoms in disaster survivors [23] and Syrian refugees [24]. CSE has been found to be a mediator for HCWs between acute stress responses and the development of PTSD in the context of disasters such as the COVID-19 pandemic [25]. Individuals with a higher sense of CSE should be less distressed by stress caused by traumatic events [9]; therefore, CSE should be inversely related to PTSD symptoms in HCWs. To date, no study has investigated the relationship between PTSD symptoms and trauma-related CSE among Greek HCWs who work with refugees.

Some HCWs who maintain positive psychological adaptation during or after assisting refugees may be referred to as “resilient” [26]. In this study, we define resilience according to Masten’s [27] definition: “the capacity of a dynamic system to adapt successfully to disturbances that threaten system function, viability, or development” (p. 6). While CSE is a perceived ability to cope, resilience is one’s capacity to maintain or develop social, psychological, or physical resources necessary for positive psychological adaptation [24,28]. The resilience ability of HCWs may be influenced by the work environment, including exposure to potentially traumatic events. Past research has found that nurses with high levels of resilience are less likely to experience PTSD compared with those with low levels of resilience [29,30]. Although some research supports the influence of resilience on PTSD among HCWs [29], the relationship between resilience and PTSD symptoms in HCWs who work with refugee populations has not been explored.

The aim of the present paper is to investigate the relationship between symptoms of PTSD and two potential predictors of adaptation to working with refugees who have experienced potentially traumatic events, trauma-related CSE, and resilience. Based on previous research conducted on refugee populations, we hypothesized that fewer symptoms of PTS would be associated with higher levels of trauma-related CSE and resilience among HCWs who work with refugees.

## 2. Materials and Methods

### 2.1. Participants

Participants were recruited primarily through an email announcement about the study from the second author (ST) sent to refugee organizations and NGOs. Utilizing snowball sampling methods, we contacted a total of 150 HCWs and gave information about the study. Snowball sampling is a non-probability sampling method where individuals may recruit future participants from their networks [31]. Inclusion criteria for the study required participants to (1) be HCWs who had worked with refugees and/or asylum seekers for at least 1 month, (2) self-identify as being exposed to potentially traumatic events (including trauma narratives from patients), (3) be residing in Greece at the time of the study, and (4) be 18 years of age or older. To assess if the participant met the inclusion criterium of being exposed to an index trauma, the researchers asked potential participants the following question: “in the past month did you experience repeated or extreme exposure to aversive details of a traumatic event(s)?”. Researchers also described common traumatic events such as: war, terrorism, sexual assault, death threats, and domestic violence as examples of traumatic events. If the participants said “yes”, they were deemed to have been exposed to a potentially traumatic event and were included in the study; however, the researchers did not document the specific index trauma for each participant. The participants were asked to reflect on the potentially traumatic experience while completing the questionnaires for the study. Only 132 people met the inclusion criteria and consented to participate in the study, and only 112 people completed the questionnaire in its entirety. Therefore, the final sample was composed of 112 HCWs.

### 2.2. Procedures

The current study was approved by the California State University Long Beach Institutional Review Board. Data were collected by the study authors in two areas of Greece, namely Athens and Crete. Recruitment occurred from March 2020 to May 2022. All participants were given a choice to complete the questionnaire in English or Greek; nonetheless, all chose to complete them in Greek. Participants received an incentive of a €5 gift card for a local supermarket for completing the questionnaires. The third author (PC) collected all quantitative data.

### 2.3. Measures

Data on age, months working with refugees and/or asylum seekers, relationship status, education, and professional affiliation were gathered using a sociodemographic questionnaire developed by the authors.

To assess PTS symptoms, the PTSD Checklist for DSM-5 (PCL-5) [32] was utilized. This is a 20-item self-report questionnaire that produces a total score and cut-off score. Participants reporting symptom scores of 33 or more were characterized as meeting criteria for probable PTSD [33]. Two Greek–English researchers translated the PCL-5 instrument from English to Greek. The researchers worked collaboratively to compare translations until they agreed on a final translated version of the instrument. A third researcher back-translated the instrument into English. Responses were rated on a five-point scale which indicated the level of distress based on trauma symptoms in the past month from 0 (not at all) to 5 (extremely)). In the current sample, PCL-5 total scores showed high internal reliability (Cronbach’s α = 0.94; N = 112) for the Greek version of the questionnaire.

For the assessment of resilience traits, we applied the Resilience Scale-11 (RS-11) [34], which is an 11-item resilience questionnaire. Five core personality characteristics of resilience are assessed by the questionnaire: (1) purpose, (2) perseverance, (3) self-reliance, (4) existential aloneness, and (5) equanimity. A seven-point scale was used to rate responses from 1 (no, I strongly disagree) to 7 (yes, I strongly agree). Two Greek–English researchers translated the RS-11 instrument from English to Greek. The researchers worked collaboratively to compare translations until they agreed on a final translated version of the instrument. A third researcher back-translated the instrument into English. The Greek version of the RS-11 showed good internal reliability (Cronbach’s α = 0.87; N = 112).

To assess perceptions of trauma-related CSE, we administered the Coping Self-Efficacy for Trauma scale (CSE-T) [35]. This nine-item questionnaire uses a seven-point scale (1 (I am not at all capable) to 7 (I’m totally capable)) to assess trauma CSE. Two researchers worked collaboratively to compare translations until they agreed on a final translated version of the instrument. A third researcher back-translated the instrument into English. The Greek version of the CSE-T showed good internal reliability (Cronbach’s α = 0.90, N = 112).

### 2.4. Data Analysis

Descriptive statistics of different variables are presented as means and standard deviations or as frequencies and percentages (Table 1). Chi-square and *t*-tests were performed to examine the associations between probable PTSD (yes/no) and sociodemographic variables. The associations among resilience, CSE, and probable PTSD variables were examined by binary logistic regression analysis. Before the analysis, the data were tested for normality of distribution in relation to both independent and dependent variables and checked for outliers. Adjusted odds ratios were calculated in order to account for significant sociodemographic characteristics. SPSS version 26.0 was utilized to analyze the data.

## 3. Results

### 3.1. Participant’s Characteristics

A total of 112 adults participated in the study (see Table 1). Among the participants, most self-identified with a professional affiliation of psychology (34.8%). Females accounted for 75% of the participants, and 70.5% of participants self-identified as single/married but not living together/divorced/widowed. The sample was relatively young, with a mean age of 31.8 years (*SD* = 7.41). The majority of participants in the sample had worked with refugees for several years, with a mean of 31.6 months working with refugees in Greece (*SD* = 25.5).

### 3.2. Sociodemographic Factors Associated with PTSD in HCWs

Probable PTSD was identified in a total of 17.9% (*n* = 20) of the participants. The mean score for probable PTSD symptoms was 19.2 (*SD* = 14.4), with a range from 0 to 62. No differences were found between the screen positive probable PTSD group compared to no probable PTSD group on any sociodemographic characteristic except for age (*t*(110) = 4.00, *p* < 0.05; see Table 1).

### 3.3. Adaptation Factors Associated with PTSD in HCWs

Correlational analysis (Pearson’s r) for the continuous variables measuring resilience, trauma-related CSE, and probable PTS symptoms showed that lower levels of current probable PTS symptoms were significantly correlated with higher trauma-related CSE ratings (*r* = −0.49, *p* < 0.001) and higher resilience ratings (*r* = −0.59, *p* < 0.01). A significant correlation between resilience and trauma-related CSE was also found (*r* = 0.70, *p* < 0.001).

In order to identify if CSE and resilience were associated with probable PTSD, we ran a logistic regression model. Table 2 shows binary logistic regression results for the probable PTSD dependent variables (Wald = 3.70, df = 1, *p* < 0.05; *B* = 3.42, S.E. = 1.78). The results indicated that a lower trauma-related CSE rating was an independent risk factor for probable PTSD in unadjusted models (Table 2; OR = 0.89, 95% CI, 0.82, 0.96, *p* < 0.01) and adjusted models (OR = 0.90, 95% CI, 0.83, 0.97, *p* < 0.01). Resilience rating was not found to be significantly associated with probable PTSD.

## 4. Discussion

To the best of our knowledge, this is the first study to examine associations among probable PTSD, trauma-related CSE, and resilience in HCWs who work with refugees and asylum seeker populations in Greece. The present study documents the presence of probable PTSD symptoms in local Greek HCWs who work with NGOs on short-term contracts. The results of the study should be cautiously interpreted with the understanding of the current sociopolitical conditions in Greece. The Greek healthcare system has undergone change and transition due to the many years of austerity in Greece, the COVID-19 pandemic, and the ongoing refugee and migration crisis [36]. These ongoing difficulties may have impacted the results of the current study and future studies are necessary to replicate and further understand risk and resilience factors in Greek HCWs.

The current study suggests that 17.9% of the sample of Greek HCWs screened positive for probable PTSD. This finding is consistent with two other studies of Greek HCWs conducted during the COVID-19 pandemic which found probable PTSD in 16.7% [36] and 15.2% [37] of the Greek study populations, respectively. The prevalence of probable PTSD found in the current study was slightly lower than a recent review of mental health conditions in HCWs during and after a pandemic, which found PTSD to be the most common mental health condition in multiple countries, with an estimated 21.7% prevalence [38]. Another review of COVID-19-related traumatic stress among HCWs also identified PTS symptoms with a prevalence ranging from 7.4% to 35% [39]. Probable PTSD in HCWs has also been found in Guatemalan aid workers [40] as well as in aid workers assisting Iraqi refugees in Jordan [41], suggesting that HCWs may be a population that needs monitoring of mental health issues and support as necessary. Significant differences between the PTSD and no PTSD groups emerged only with respect to age, with the PTSD group being younger in age. This is consistent with the literature, which suggests that younger HCWs are more likely to experience mental distress [42]. Our study stipulates that a younger age may be a risk factor for developing probable PTSD symptoms in HCWs, thus underlining the importance of adequate training and support for young professionals who become HCWs for vulnerable populations. Perhaps, identifying risk factors, such as lower CSE, in young HCWs may assist supervisors in developing a training plan aimed at supporting wellbeing. It is important to adequately screen HCWs for trauma exposure and distress and to address their needs in order to prevent emerging, ongoing, or chronic mental health issues in this population. Two recent reviews recommended the inclusion of self-care strategies as part of both on-boarding and support of HCWs [43,44]. These include strategies such as relaxation techniques, using e-mental health services, prioritizing a healthy lifestyle that includes adequate sleep, supportive relationships, activities and hobbies, and mindfulness meditation [45]. Supervision, mutual support opportunities, and self-care are vital components of care work [46] and can enable the development of good practice or prevent mental health difficulties in such challenging work settings.

The quantitative results indicate that lower levels of PTSD symptoms were significantly correlated with higher levels of trauma-related CSE and resilience among HCWs who work with refugees. This is consistent with research that shows that CSE is inversely associated with a person’s vulnerability to psychological issues [47] and, specifically, PTSD [23,24]. We found that lower trauma-related CSE was predictive of probable PTSD; however, resilience ratings were not predictive of probable PTSD. Our results are consistent with social cognitive theory [21]. Social cognitive theory suggests that individuals with higher CSE may be better able to adapt to indirect and/or direct trauma exposure, something which may prevent the development of PTSD symptoms [21]. Considering the inverse relationship between trauma CSE and PTSD symptoms found in the current population of Greek HCWs, additional training for HCWs may be necessary to further strengthen trauma-related CSE. For example, training to assist with the development of specific skills and knowledge to enhance a sense of personal control to manage traumatic event exposure may be warranted, as these skills may promote psychological adaptation. For example, training in mastery modeling (i.e., self-defense training) may be one skillset used to enhance self-assertive action [48]. The resilience measure used in this study did not characterize resilience as resources necessary for adaptation to stressful, difficult, or dangerous stimuli, but instead defined resilience as a personality characteristic. Perhaps, this was the reason why lower resilience was not found to be predictive of probable PTSD. Future studies using other definitions of resilience are necessary to further investigate this relationship.

The findings of the current study should be interpreted in light of some important limitations. The results of the study were gathered from Greek HCWs who specifically worked with refugees and asylum seekers and cannot be generalized to other HCW populations. The current study used a single question to assess trauma exposure and, if the participant self-reported experiencing or witnessing any traumatic event, they were deemed to have met the inclusion criteria. A formal assessment of PTSD Criterion A was not done; therefore, throughout the current study, the authors stated they assessed probable PTSD or PTS symptoms [49]. There was no assessment of potentially traumatic events experienced over the course of their lifetime or follow-up interviews; therefore, pre-existing PTSD, possible delayed onset PTSD, or recovery from symptoms was not accounted for when designing the study or in the analysis of the results. Because the study was cross-sectional, the results cannot reveal causality, due to a lack of established temporality, or the course of probable PTSD in HCWs. Although all HCWs in the current study reported working with refugees who had experienced traumatic events, it is not known whether the HCWs experienced other traumatic event exposure during the pandemic. Another limitation is the use of self-report measures instead of a standardized clinical interview to assess PTSD. Also, the effect of resilience may have been impeded in the models by CSE. This study gathered data from HCWs between 2020 and 2022. The study was implemented during the COVID-19 pandemic and issues associated with the pandemic may have impacted the results.

## 5. Conclusions

In conclusion, the current study reveals that HCWs working with refugees and/or asylum seekers in Greece may be at risk of probable PTSD; however, higher trauma-related CSE may prevent the development of probable PTSD in some HCWs. A substantial body of research indicates that the majority of individuals exposed to potentially traumatic events do not develop PTS symptoms or, in fact, the most acute symptoms of PTSD [50]. Healthcare workers, because of the nature of caring and providing direct services to victims of horrific events, may be at greater risk of developing distress and psychopathologies. Our study findings highlight the need to provide screening, assessment, training, and mental health support for HCWs, particularly younger HCWs who work with refugees and asylum seekers. The findings suggest that a focus on CSE in training could assist HCWs. Self-efficacy may be enhanced by training modules focused on coping skills in response to witnessing or experiencing traumatic events, as well as by ways to enhance a sense of agency. Additional studies are necessary to understand the long-term implications of working with refugee and asylum seeker populations and the type and amount of training and support necessary to assist HCWs with positive psychological adaptation and self-care.

## Figures and Tables

**Table 1 behavsci-14-00509-t001:** Relationship between sociodemographic characteristics and probable PTSD (N = 112).

Characteristics	Whole Sample (N = 112)	No Probable PTSD (*n* = 92)	Probable PTSD (*n* = 20)	Difference between Groups	*p*
Age *M* (*SD*) *22–62 years age range*	31.8 (7.41)	32.41(7.90)	28.8 (3.40)	t(110) = 4.00	0.048
Months working with refugees *M* (*SD*) *1.5–156 months range*	31.6 (25.5)	33.12 (27.07)	24.95 (15.30)	t(110) = 1.69	0.20
Gender *n* (*%*)				*χ*^2^(1) = 1.30	0.39
Female	84 (75)	67 (72.8)	17 (85.0)		
Male	28 (25)	25 (27.2)	3 (15.0)		
Relationship Status *n* (*%*)				*χ*^2^(1) = 0.00	1.00
Single/Married but not living together/divorced/widowed	79 (70.5)	65 (70.7)	14 (70.0)		
Married or living together	33 (29.3)	27 (29.3)	6 (30.0)		
Education *n* (*%*)				*χ*^2^(2) = 0.66	0.72
PhD or MD,	11(9.8)	10 (10.9)	1 (5.0)		
Master’s degree	41(36.6)	33 (35.9)	8 (30.0)		
BS degree	60(53.6)	49 (53.3)	11 (40.0)		
Professional affiliation *n* (*%*)				*χ*^2^(4) = 0.81	0.94
Medicine	11(9.8)	10 (10.9)	1 (5.0)		
Psychology	39 (34.8)	32 (34.8)	7 (35.0)		
Social Work	33 (29.3)	27 29.3)	6 (30.0)		
Nursing	11(9.8)	9 (9.8)	2 (10.0)		
Other	18 (16.1)	14 (15.2)	4 (20.0)		

**Table 2 behavsci-14-00509-t002:** Binary logistic regression to identify significant predictors of PTSD (N = 112).

	Whole Sample (N = 112)	Probable PTSD ^a^ (*n* = 20)	No Probable PTSD (*n* = 92)	Crude OR (95% CI)	Adjusted OR ^d^ (95% CI)
RS-11 ^b^ *M* (*SD*)	59.79 (9.52)	55.30 (7.74)	60.76 (9.62)	1.01 (0.94–1.08)	1.01 (0.94–1.09)
CSE-T ^c^ *M* (*SD*)	48.13 (8.96)	41.25 (6.45)	49.62 (8.75)	0.89 (0.82–0.96) **	0.90 (0.83–0.97) **

^a^ PTSD Checklist for DSM-5 (PCL-5). ^b^ Resilience Scale-11. ^c^ Trauma Coping Self-Efficacy (CSE-T). ^d^ Adjusted for significant variables (age). ** *p* < 0.01.

## Data Availability

Data will be available upon request.

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
