# Peer review of "Resilience, Coping Self-Efficacy, and Posttraumatic Stress Symptoms among Healthcare Workers Who Work with Refugees and Asylum Seekers in Greece"

_behavsci, 2024, doi:10.3390/bs14060509_

Round 1
Reviewer 1 Report
Comments and Suggestions for Authors
Review
This is a paper about a little studied but likely under-duress group of providers—HCWs caring for refugee populations. The focus was on understanding, in a cross sectional study, whether coping self efficacy and resilience are related to PTSD. The findings show that in this sample of HCWs, ~17% had probable PTSD. Higher CSE reduced odds of PTSD. Being younger was also associated with PTSD. Resilience was not. The analysis was somewhat weak and thin. The effect sizes are not reported, which would improve the reader’s ability to know the strength of the association.
That said, the population is not recruit, and the area of research is both important and novel. I believe this does make a contribution. It would have been ideal to have a more complex model that included more known covariates.
The introduction and discussion would benefit from doing a more in-depth literature search that looks at self-efficacy and trauma outcomes in healthcare workers, whether those HCWs work in refugee populations or not.
The authors seem to get away with a thin literature review by stating that no one has ever studied HCWs working with refugees. But a lot of people have studied HCWs working in stressful conditions…this (refugee populations) being one of them. So, a more expansive and scoping lit review that seeks to cite recent or leading research on PTSD and self efficacy among HCWs in other settings would be of value if this paper wants to add to the lit and be seen as a credible contribution.
It’s a reasonable contribution with some improvements.
Intro
There is a significant lit on HCWs in stressful contexts that is almost entirely avoided in the lit review of this paper. Its not enough to say “no one has ever studied HCWs working with refugees on concepts of PTSD and resilience and CSE.” That’s like saying no one has ever studied the nutritional value of purple carrots, so therefore, we don’t know anything about them.” Orange carrots are probably a pretty fair parallel to purple carrots.
Your intro may be improved if you give some examples, just phenomenologically, of some of the scary, traumatic, unsolvable crises that HCWs working with refugees may be encountering.
In the 4th paragraph in the introduction, it may benefit your arugment to cite work showing that coping self-efficacy is a mediator for healthcare workers between acute stress responses and the development of PTSD—in other disaster contexts (i.e., pandemic).
Smith, A. J., Shoji, K., Griffin, B. J., Sippel, L. M., Dworkin, E. R., Wright, H. M., ... & Benight, C. C. (2022). Social cognitive mechanisms in healthcare worker resilience across time during the pandemic. Social psychiatry and psychiatric epidemiology, 57(7), 1457-1468.
Results
Provide range of PTSD scores in your sample.
Report your logistic regression findings with effect sizes to couch in this in a way that readers can understand how large the effect is.
The effect of resilience was likely washed out in the models by CSE (see the collinearity between Resil and CSE… r = .7). This probably means that you didn’t really study whether resilience increases or decreases risk for PTSD diagnosis, because the measure with the most variability (CSE) won the race being that both measures measure almost the same thing. You may want to talk about this in the limitations.
Discussion
Compare your PTSD rates in your sample of HCWs to HCWs in other contexts beyond Greece, for context.
I think this is actually one of the most key findings for this study, even though it wasn’t your “variable of interest”: Our 201 study stipulates that younger age may be a risk factor for developing PTSD symptoms in 202 HCWs, thus underlining the importance of adequate training and support for young pro-203 fessionals who become HCWs for vulnerable populations.”
Any way to magnify and talk more about this? This seems like a real tangible piece of wisdom that can come out of this. How do we protect our younger HCWs who are virtuous and ready to go in to combat? Can we identify those that have lower CSE and higher risk factors, and titrate the extent to which we throw them in the deep end while we build up their efficacy?
Author Response
Response to Reviewer 1:
Intro
Reviewer Comment: There is a significant lit on HCWs in stressful contexts that is almost entirely avoided in the lit review of this paper. Its not enough to say “no one has ever studied HCWs working with refugees on concepts of PTSD and resilience and CSE.” That’s like saying no one has ever studied the nutritional value of purple carrots, so therefore, we don’t know anything about them.” Orange carrots are probably a pretty fair parallel to purple carrots.
Your intro may be improved if you give some examples, just phenomenologically, of some of the scary, traumatic, unsolvable crises that HCWs working with refugees may be encountering.
Response: Thank you for your helpful suggestions. We have added an additional sentence summarizing additional literature citing exposure to potentially traumatic events to paragraph 3 of the introduction. We have carefully reviewed the introduction and added some examples of traumatic and stressful life experiences that HCWs experience that have been documented in the literature in paragraph 2 of the introduction.
In the 4th paragraph in the introduction, it may benefit your arugment to cite work showing that coping self-efficacy is a mediator for healthcare workers between acute stress responses and the development of PTSD—in other disaster contexts (i.e., pandemic).
Smith, A. J., Shoji, K., Griffin, B. J., Sippel, L. M., Dworkin, E. R., Wright, H. M., ... & Benight, C. C. (2022). Social cognitive mechanisms in healthcare worker resilience across time during the pandemic. Social psychiatry and psychiatric epidemiology, 57(7), 1457-1468.
Response: We have added a sentence in the 4th paragraph stating that CSE has been found to be a mediator for HCWs between acute stress response and the development of PTSD.
Results
Provide range of PTSD scores in your sample.
Response: We have added the range of probable PTSD symptom scores in the Results section 2.4.
Report your logistic regression findings with effect sizes to couch in this in a way that readers can understand how large the effect is.
Response: In the logistic regression model, odds ratio can be used as an effect size statistic. We have included odds ratios as well as confidence intervals in Table 2. Also, in the Results section we added additional information from the logistic regression analysis. We are happy to add additional information if the Reviewer thinks it will improve the manuscript.
The effect of resilience was likely washed out in the models by CSE (see the collinearity between Resil and CSE… r = .7). This probably means that you didn’t really study whether resilience increases or decreases risk for PTSD diagnosis, because the measure with the most variability (CSE) won the race being that both measures measure almost the same thing. You may want to talk about this in the limitations.
Response: We have added this information to the limitations section
Discussion
Compare your PTSD rates in your sample of HCWs to HCWs in other contexts beyond Greece, for context.
Response: We have added this information to paragraph 2 of the Discussion section.
I think this is actually one of the most key findings for this study, even though it wasn’t your “variable of interest”: Our 201 study stipulates that younger age may be a risk factor for developing PTSD symptoms in 202 HCWs, thus underlining the importance of adequate training and support for young pro-203 fessionals who become HCWs for vulnerable populations.”
Any way to magnify and talk more about this? This seems like a real tangible piece of wisdom that can come out of this. How do we protect our younger HCWs who are virtuous and ready to go in to combat? Can we identify those that have lower CSE and higher risk factors, and titrate the extent to which we throw them in the deep end while we build up their efficacy?
Response: We have added information to paragraph 2 of the discussion highlighting strategies to support young healthcare workers. We are happy to add additional information as necessary.
Reviewer 2 Report
Comments and Suggestions for Authors
I thank the authors for their study on an important topic. I have however several comments.
I would advise the authors to not call it PTSD, as they did not measure PTSD. PTSD requires criterion A. This was a huge problem during COVID-19. As the problem the authors have is the same, see and cite Muysewinkel E, Stene LE, Van Deynse H, Vesentini L, Bilsen J, Van Overmeire R. Post-what stress? A review of methods of research on posttraumatic stress during COVID-19. J Anxiety Disord. 2024;102:102829. doi:10.1016/j.janxdis.2024.102829
The reason why this is problematic, is because the authors are actually going against the prime definition of PTSD, namely the trauma indicator. Look at criterion A (again, just see the review cited above, because it explains it much clearer than I can here).
It seems that the respondents in this study were exposed in a secondary way. However, we cannot know, since there was no criterion A indications. Again, this is severe limitation to the study, which needs to be acknowledged.
Related to that, the author speak of “trauma related” CSE; yet do not assess trauma. Furthermore, they state “This 9-item questionnaire uses a 7-point scale (1 (I am not at all capable) to 7 (I’m totally capable)) to assess trauma”. How does it “assess” trauma?
Data collection also took 2 years?
The inclusion was self-reported trauma exposure. How was this measured? How was this decided? What was defined as trauma? Did the respondents decide what was trauma or did the researchers provide a list of what trauma is?
What is “self-identification as 18 years or older”? I suspect that is just means “18 years or older”?
In short: the authors need to use a better conceptualization of PTSD, PTSS and trauma. This will have an impact on the introduction, methods and discussion, as they will need to be seriously rewritten with in mind the definition of PTSD, and then more precisely, what is the authors actually measured, namely stress.
Author Response
Response to Reviewer 2:
I would advise the authors to not call it PTSD, as they did not measure PTSD. PTSD requires criterion A. This was a huge problem during COVID-19. As the problem the authors have is the same, see and cite Muysewinkel E, Stene LE, Van Deynse H, Vesentini L, Bilsen J, Van Overmeire R. Post-what stress? A review of methods of research on posttraumatic stress during COVID-19. J Anxiety Disord. 2024;102:102829. doi:10.1016/j.janxdis.2024.102829
The reason why this is problematic, is because the authors are actually going against the prime definition of PTSD, namely the trauma indicator. Look at criterion A (again, just see the review cited above, because it explains it much clearer than I can here).
It seems that the respondents in this study were exposed in a secondary way. However, we cannot know, since there was no criterion A indications. Again, this is severe limitation to the study, which needs to be acknowledged.
Response: We thank the Reviewer for bringing this important point to our attention, and we acknowledge that we did not do a formal assessment of PTSD Criterion A for the current study. We have noted this as a limitation in the last paragraph of the Discussion section. Moreover, in the Participants section, 2.1., we added information regarding how we assessed whether the Participant met the inclusion criteria and if they reported that they experienced an index trauma by a single question. We added the following details to the manuscript ““in the past month did you experience repeated or extreme exposure to aversive details of a traumatic event(s)?” Researchers also described common traumatic events such as: war, terrorism, sexual assault, threatened death, and domestic violence as examples of traumatic events. If the participants said “yes” they were deemed to have been exposed to a potentially traumatic event and were included in the study, however, the researchers did not document the specific index trauma for each participant. The participants were asked to reflect on the potentially traumatic experience while completing the questionnaires for the study.”
Related to that, the author speak of “trauma related” CSE; yet do not assess trauma. Furthermore, they state “This 9-item questionnaire uses a 7-point scale (1 (I am not at all capable) to 7 (I’m totally capable)) to assess trauma”. How does it “assess” trauma?
Response: The CSE-trauma scale has the following instructions for Participants: “This assessment is designed to have you think about important issues related to the dealing with exposure to a traumatic event. For each of the situations described below, you are asked to rate how capable you are that you can successfully deal with them. Because people differ from each other in the way that they are dealing with this type of event there is no single correct response. Please think about yourself currently not as it was the day of the traumatic experience.” As mentioned above, the researchers asked a single question to assess trauma exposure, and participants were asked to reflect on the traumatic event identified when answering the questionnaires.
Data collection also took 2 years?
Response: Yes, it took 2 years due to the COVID-19 pandemic. Data collection began shortly after the pandemic started in 2020.
The inclusion was self-reported trauma exposure. How was this measured? How was this decided? What was defined as trauma? Did the respondents decide what was trauma or did the researchers provide a list of what trauma is?
Response: Please see above response. Trauma exposure was measured with a single question.
What is “self-identification as 18 years or older”? I suspect that is just means “18 years or older”?
Response: Yes, we changed the manuscript to “18 years or older”.
In short: the authors need to use a better conceptualization of PTSD, PTSS and trauma. This will have an impact on the introduction, methods and discussion, as they will need to be seriously rewritten with in mind the definition of PTSD, and then more precisely, what is the authors actually measured, namely stress.
Response: We agree with the Reviewer that the measure of trauma exposure was insufficient. We added information throughout the manuscript suggesting we measured PTS symptoms and probable PTSD. In addition, we added details to how we measured trauma exposure. Finally, we added this as a major limitation in the Discussion section.
Round 2
Reviewer 2 Report
Comments and Suggestions for Authors
Is okay for me now. Thank you for the great revision!